# Chemo-Immunotherapy in First Line Extensive Stage Small Cell Lung Cancer (ES-SCLC): A Systematic Review and Meta-Analysis

**Arani Sathiyapalan \*, Michela Febbraro, Gregory R. Pond and Peter M. Ellis**

Department of Oncology, Juravinski Cancer Centre, McMaster University, Hamilton, ON L8V 5C2, Canada
\* Correspondence: sathiyapal@hhsc.ca

**Abstract:** Small cell lung cancer (SCLC) is an aggressive neuroendocrine carcinoma with early metastatic potential. The standard-of-care treatment has not changed in years. Recent studies report improved progression-free survival (PFS) and overall survival (OS) with combined ICI and chemotherapy in ES-SCLC. We conducted a systematic review and meta-analysis to assess the magnitude of survival benefits. We searched MEDLINE, EMBASE, and Cochrane between 1 January 2010 and 15 July 2022 and conference proceedings from 2018 to 2022, for randomised controlled trials, evaluating chemo-ICI compared with platinum-doublet chemotherapy in untreated ES-SCLC. Outcomes assessed were PFS, OS, objective response rate (ORR), duration of response (DoR), toxicity, and health-related quality of life (HRQoL). The search identified 8061 studies, with 8 (56 publications) included in the final analysis. PFS and OS were significantly improved for patients randomised to chemo-ICI (PFS hazard ratio (HR) 0.75, 95% confidence interval (CI) 0.70–0.80) and (OS HR 0.79, 95% CI 0.73–0.85). Subgroup analysis demonstrated a differential effect between PD-1/PD-L1 and CTLA-4 inhibitors. There was no difference in ORR and DoR. All-grade adverse events (RR 1.06, 95% CI 1.00–1.12) were similar. The addition of ICI to chemotherapy in untreated ES-SCLC results in a 22% risk reduction in death, and a 25% risk reduction in disease progression with a minimal increase in toxicity. These improvements are modest but represent progress beyond the standard of care.

**Keywords:** SCLC; chemotherapy; immunotherapy; extensive stage; chemo-immunotherapy

## 1. Introduction

Small cell lung cancer is a highly aggressive neuroendocrine tumour arising from the lung. It has a predilection for rapid growth and early metastatic potential with the majority of patients diagnosed at an advanced stage termed extensive stage small cell lung cancer (ES-SCLC) [1]. The current systemic treatment regimen consisting of a platinum agent and etoposide chemotherapy has been the standard of care for approximately 20–30 years with high initial response rates of 60–65%, and with relatively short time to its progression. Median overall survival (OS) is approximately 10 months with fewer than 15% of patients living beyond two years [2,3]. New therapeutic regimens have unfortunately failed to improve survival or change the overall prognosis of ES-SCLC [4].

SCLC is strongly associated with smoking and is a carcinogen-associated tumour with one of the highest rates of tumour mutations per mega-base [5,6]. Mutations in tumour DNA can give rise to neoantigens that are recognised and targeted by the immune system [7]. The more mutations a tumour has, the more neoantigens it is likely to form. Immune checkpoint inhibitors (ICI) including programmed death-1 (PD-1) inhibitors, programmed death ligand-1 (PD-L1) inhibitors, and cytotoxic T-lymphocyte associated protein 4 (CTLA-4) inhibitors promote immune-mediated targeted killing of tumour cells. Tumour mutation burden (TMB) is useful in estimating tumour neoantigen load [8] and is consequently investigated as a predictive marker of response to ICI. Cancers associated with high TMB including melanoma, lung, and colorectal cancers and have been found

to be more likely to respond to ICI [8,9]. In non-small cell lung cancer (NSCLC), ICI are now routinely incorporated into treatment algorithms with the demonstrated superior OS. ICI in combination with platinum-based chemotherapy have recently been investigated based on the hypothesis of initial cytotoxic damage by chemotherapy releasing tumour-associated antigens, which, when timed concurrently with ICI, can synergistically amplify T-cell priming and subsequent immune-mediated anti-tumour responses. This combination has also demonstrated superior OS to platinum-based chemotherapy in multiple trials of first-line therapy for NSCLC [10,11] with reasonable tolerability.

Similar to NSCLC, there is also a suggestion of the benefit of ICI in ES-SCLC. In line with TMB data in other cancers, an improvement in objective response rate (ORR) and OS was observed in patients with high TMB status [12]. Furthermore, several trials have evaluated the addition of an ICI to platinum-based chemotherapy in ES-SCLC, to determine if the results observed from chemo-ICI in NSCLC can be replicated in SCLC [13–15]. Currently, the magnitude of benefit from the addition of ICI to chemotherapy, as well as overall tolerability is unclear, and a meta-analysis of the available data is required to aid in clinical decision-making.

The objective of this systematic review and meta-analysis is to assess the magnitude of efficacy and tolerability of combined chemo-ICI in the first-line systemic treatment of ES-SCLC as compared with standard-of-care chemotherapy.

## 2. Materials and Methods

### 2.1. Protocol and Registration

The protocol was registered with the International Prospective Register of Systematic Reviews (PROSPERO no.: CRD42020189779) and followed the 2009 PRISMA checklist for systematic reviews and meta-analysis.

### 2.2. Search Strategy

An electronic search was undertaken of MEDLINE, EMBASE, and the Cochrane Central Register of Controlled Trials (CENTRAL) (January 2010–15 July 2022) (Appendix A). The search was restricted to start in 2010 as ICI were first reported in the literature in 2012 [16]. A targeted search of the grey literature, including ClinicalTrials.gov and abstracts from relevant conference proceedings (American Society of Clinical Oncology (ASCO) meeting, European Society for Medical Oncology (ESMO) meeting, World Conference on Lung Cancer (WCLC)) was also performed. References of selected trials were reviewed to identify additional relevant studies. There were no restrictions on language or publication status.

### 2.3. Eligibility

Studies were included if they met the following criteria:

- Patients with newly diagnosed ES-SCLC with no previous systemic treatment;
- Therapy with combined chemo-immunotherapy with any of the following ICI:
  - PD-1 inhibitors;
  - PD-L1 inhibitors;
  - CTLA4 inhibitors.
- Comparator arm is standard of care platinum-based chemotherapy doublet including:
  - Cisplatin or Carboplatin and Etoposide;
  - Cisplatin or Carboplatin and Paclitaxel
- Randomised controlled trials.

Studies were excluded if treatment was in limited stage (LS) SCLC or in the second or subsequent lines of therapy. Non-randomised studies such as retrospective trials were also excluded.

### 2.4. Study Selection

Titles and abstracts of identified studies were transferred into Covidence [17] and evaluated by two independent reviewers (AS and MF). Studies meeting inclusion criteria

were advanced for full-text review. Disagreements were resolved through discussion and if necessary, were resolved after discussion with a third author (PME).

### 2.5. Risk of Bias and Data Extraction

Two review authors (AS and MF) independently assessed the risk of bias using the recommendations in the Cochrane Handbook for Systematic Reviews of Interventions (Risk of Bias Tool 2.0) [18]. For each included study, two review authors (AS and MF) independently extracted data into Covidence. Any differences in data extraction were resolved after a discussion with a third author (PME).

### 2.6. Data Items

Data items extracted included study characteristics, publication type, study participant characteristics, intervention characteristics, and comparator characteristics. Outcome data of interest included PFS, OS, ORR, and DoR. Safety and tolerability data were extracted as rates of Grade 1–4 and Grade 3–4 adverse events for toxicities of clinical interest (anemia, thrombocytopenia, neutropenia, febrile neutropenia, fatigue, nausea, and vomiting). Grade 5 adverse event rates were also collected. Immune-related adverse events (iRAE) were extracted separately from the chemo-ICI arms as overall Grade 1–4 and Grade 3–4 adverse events, and individual rates for toxicities of clinical interest (rash, colitis, pneumonitis, hepatitis, and nephritis). Health-related quality of life (HRQoL) data as measured by validated tools in lung cancer were extracted if reported.

### 2.7. Summary Measures and Synthesis of Results

Raw data were entered into Review Manager (RevMan) 5.0 [19] for data analysis. Time-to-event outcomes (PFS and OS) were extracted as median point estimates, hazard ratios (HR), and 95% confidence intervals (CI). The generic inverse variance method was used to pool hazard ratios from the independent studies. Dichotomous outcomes (ORR and toxicity) were extracted as event counts and denominators. ORR was extracted as the number of patients experiencing partial or complete responses in the intervention group and the comparator group. The fixed effect Mantel–Haenszel method was used to calculate the pooled risk ratios and its 95% CI. Continuous outcomes (DoR) were extracted as median point estimates and entered as a continuous outcome and using the fixed effect Mantel–Haenszel method, a pooled mean difference along with 95% CI was calculated.

### 2.8. Subgroup Analyses

Given the evidence of differences in efficacy and toxicity between PD-1/PD-L1 inhibitors and CTLA-4 inhibitors, as well as between single ICI agent therapy and combination therapy [10,20–24], a pre-specified subgroup analysis was undertaken to investigate potential heterogeneity between PD-1/PD-L1 inhibitors and CTLA-4 inhibitors. When studies included combination ICI therapy (i.e., PD-1/PD-L1 inhibitor + CTLA-4 inhibitor + chemotherapy), the treatment arm was included within the PD-1/PD-L1 subgroup so as not to overestimate the efficacy of CTLA-4 inhibitors due to PD-1/PD-L1 inhibition.

### 2.9. Assessment of Certainty of Evidence

Two authors (AS and MF) assessed the certainty of evidence using the GRADE approach [25].

## 3. Results

A total of 8061 studies and abstracts were identified in the initial search. After exclusion during screening, 31 studies remained for full-text review. There were 23 studies excluded from full-text review due to not meeting the eligibility criteria of comparator or outcomes, and data from 8 studies (56 publications) [13–15,26–31] were included in the final analysis (PRISMA flow diagram—Appendix B). Within the eight studies, there were a total of ten treatment arms with ICI. CASPIAN was a three-arm trial with two experimental arms

(Durvalumab + etoposide and platinum (EP), and Durvalumab + Tremelimumab + EP). Reck et al. (2013) also conducted a three-arm trial with two experimental arms (Concurrent Ipilimumab + carboplatin + paclitaxel and Phased Ipilimumab + carboplatin + paclitaxel).

*3.1. Study Characteristics*

Individual study characteristics are listed in Table 1. The 8 studies and 10 treatment arms included a total of 3952 participants. Survival data were available for all included studies and treatment arms (PFS, OS, ORR) except for DoR, for which only 6 studies (7 treatment arms) reported data. Toxicity data (Grade 1–4 and Grade 3–4 overall toxicity) were available from seven studies (nine treatment arms). The majority of trials evaluated a PD-1 inhibitor, or PD-L1 inhibitor concurrently during chemotherapy with ongoing maintenance ICI until disease progression. Two trials (3 treatment arms) evaluated ipilimumab (CTLA-4 inhibitor) in combination with chemotherapy. This was given either in a concurrent or phased fashion. In the concurrent regimen, ipilimumab plus carboplatin and paclitaxel were given together for four cycles, followed by two cycles of carboplatin and paclitaxel. In the phased regimen, paclitaxel and carboplatin were given initially for two cycles, followed by four cycles of ipilimumab, paclitaxel, and carboplatin. A combination of PD-L1 inhibitor (durvalumab) and CTLA-4 inhibitor (tremelimumab) was studied in 1 treatment arm.

In most of the trials, the chemotherapy was cisplatin or carboplatin with etoposide. The chemotherapy backbone in one trial (two treatment arms) was carboplatin and paclitaxel.

Participant characteristics were comparable with the majority having an ECOG performance status of 0 or 1. Prophylactic cranial irradiation was explicitly allowed in participants with complete responses in 5 studies.

**Table 1.** Table of included studies and characteristics. I, intervention; C, comparator; PCI, prophylactic cranial irradiation; HR, hazard ratio; mPFS, median progression-free survival; mOS, median overall survival; EP, etoposide platinum; Y, yes.

| First Author | Year | Title | Type of Publication | Type of Study | Total No. of Patients | | Intervention | Comparator | Main | PCI | HR for PFS | mPFS (Treatment) | mPFS (Control) | HR for OS | mOS (Treatment) | mOS (Control) |
|---|---|---|---|---|---|---|---|---|---|---|---|---|---|---|---|---|
| | | | | | I | C | | | | | | | | | | |
| Wang | 2022 | CAPSTONE-1 [26] | Publication | Phase III RCT | 230 | 232 | Adebrelimab 20 mg/kg (Cycle 1–6) EP q3w × 4–6 Carbo | EP q3w × 4–6 | Y | Y | 0.67 | 5.8 | 5.6 | 0.72 | 15.3 | 12.8 |
| Cheng | 2022 | ASTRUM-005 [27] | Abstract | Phase III RCT | 389 | 196 | Serplulimab 4.5 mg/kg × 4 EP q3w × 4 Carbo | EP q3w × 4 | Y | ? | 0.48 | 5.7 | 4.3 | 0.63 | 15.4 | 10.9 |
| Reck | 2013 | CA184-041 [28] | Publication | Phase II RCT | 43 | 45 | Concurrent Ipilimumab 10 mg/kg (Cycle 1–4) Carbo/Pacli × 6 | Carbo/Pacli | Y | ? | 0.93 | 3.89 | 5.19 | 0.95 | 9.13 | 9.92 |
| Reck | 2013 | CA184-041 [28] | Publication | Phase II RCT | 42 | 45 | Phased Ipilimumab 10 mg/kg (Cycle 3–6) Carbo/Pacli × 6 | Carbo/Pacli | Y | ? | 0.93 | 5.22 | 5.19 | 0.75 | 12.94 | 9.92 |
| Reck | 2016 | CA 184–156 [13] | Publication | Phase III RCT | 566 | 566 | Phased Ipilimumab 10 mg/kg (Cycle 3–6) EP q3w × 4 Cis or Carbo | EP q3w × 6 | Y | Y | 0.85 | 4.6 | 4.4 | 0.81 | 11 | 10.9 |
| Horn | 2018 | IMpower133 [14] | Publication | Phase III RCT | 201 | 202 | Atezolizumab 1200 mg q3w EP q3w × 4 (carbo) | EP q3w × 4 | Y | Y | 0.77 | 5.2 | 4.3 | 0.70 | 12.3 | 10.3 |
| Paz-Ares | 2019 | CASPIAN [15] | Publication | Phase III RCT | 268 | 269 | Durvalumab 1500 mg q3w EP q3w × 4 Cis or carbo | EP × 6 cycles | Y | Y | 0.80 | 5.1 | 5.4 | 0.73 | 13.0 | 10.3 |

**Table 1.** *Cont.*

| First Author | Year | Title | Type of Publication | Type of Study | Total No. of Patients | | Intervention | Comparator | Main | PCI | HR for PFS | mPFS (Treat-ment) | mPFS (Con-trol) | HR for OS | mOS (Treat-ment) | mOS (Con-trol) |
|---|---|---|---|---|---|---|---|---|---|---|---|---|---|---|---|---|
| | | | | | I | C | | | | | | | | | | |
| Goldman | 2021 | CASPIAN [29] | Publication | Phase III RCT | 268 | 269 | Durvalumab 1500 mg q3w Tremelimumab 75 mg q3w EP q3w | EP × 6 cycles | Y | Y | 0.84 | 4.9 | 5.4 | 0.82 | 10.4 | 10.5 |
| Rudin | 2020 | KEYNOTE-604 [30] | Publication | Phase III RCT | 228 | 225 | Pembro EP q3w Cis or Carbo | EP q3w × 4 cycles | Y | Y | 0.73 | 4.8 | 4.3 | 0.80 | 10.8 | 9.7 |
| Leal | 2020 | EA5161 [31] | Abstract | Phase II RCT | 80 | 80 | Nivolumab 360 mg q3w EP q3w × 4 Cis or carbo | EP q3w × 4 cycles | Y | ? | 0.65 | 5.5 | 4.6 | 0.67 | 11.3 | |

*3.2. Risk of Bias*

The overall risk of bias within studies using the RoB 2.0 tool [18] was judged to be low to moderate. The risk of bias graph and summary are shown in Appendix C.

*3.3. GRADE Assessment*

The summary of findings table was prepared with GRADEpro software [32] and reported in Appendix D.

*3.4. Effects of Interventions*

3.4.1. Progression-Free Survival

PFS was available from all included studies. PFS in the chemo-ICI arms ranged from 3.9–5.8 months (median PFS 5.15m) and PFS in the comparator arms ranged from 4.3–5.6 months (median PFS 4.5m). Pooled analysis revealed significantly improved PFS for patients randomised to chemo-ICI compared with chemotherapy alone with an HR of 0.75, 95% CI 0.70–0.80 (Figure 1). With a reported 6-month PFS rate of 23.8% in the comparator group, the calculated absolute effect was 103 more patients without progression at 6 months per 1000 (95% CI 79 to 128 more). There is high certainty that chemo-ICI results in a moderate improvement in PFS in first-line ES-SCLC. The pooled analysis of PD-1/PD-L1 inhibitors revealed an HR of 0.71 and a 95% CI of 0.66–0.77, whereas the pooled HR for PFS using CTLA-4 inhibitors was 0.85 with a 95% CI of 0.76–0.96. This prespecified subgroup analysis demonstrated a significant differential effect between PD-1/PD-L1 inhibitors and CTLA-4 inhibitors.

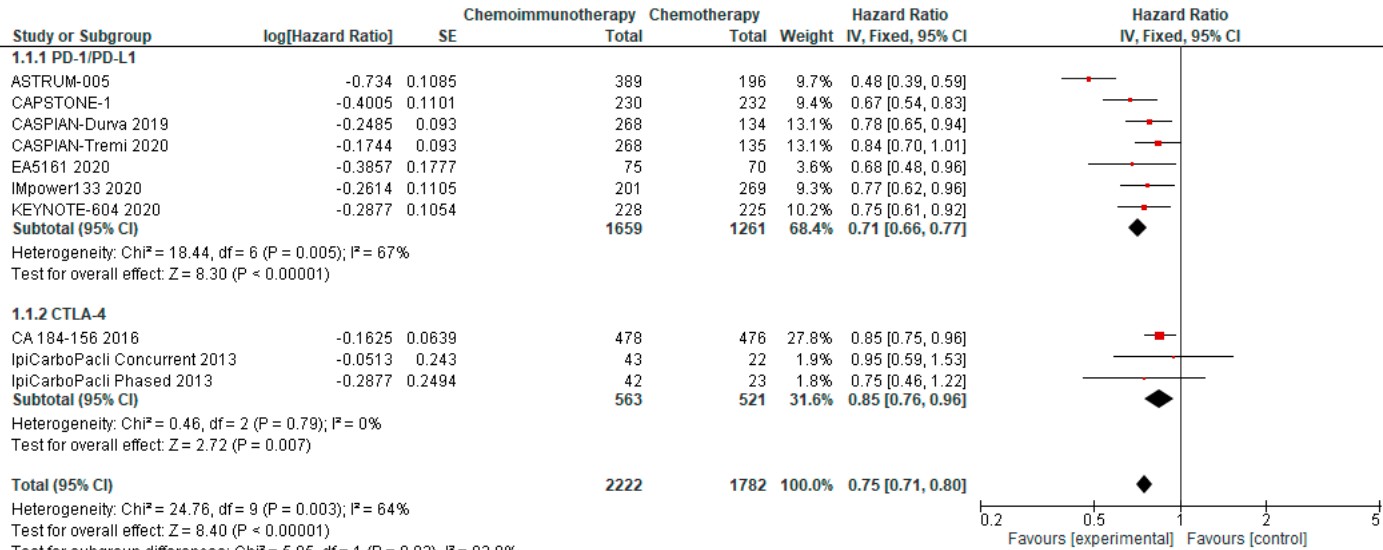

**Figure 1.** Forest Plot of hazard ratios for progression-free survival of patients with ES-SCLC and treated with chemo-ICI versus standard chemotherapy. PD–1/PD–L1, programmed death (ligand)–1; CTLA–4, cytotoxic T-lymphocyte-associated protein 4; HR, hazard ratio; CI, confidence interval; IV, inverse variance.

3.4.2. Overall Survival

OS was available from all included studies and treatment arms. On pooled analysis, OS was significantly improved with combined chemo-ICI with an HR of 0.79, 95% CI 0.73–0.85 (Figure 2). Median overall survival ranged from 9.1–15.4 months in the treatment arms as compared with a range of 8.5–12.8 months in the control arms. With a reported 12-month survival rate of 40% in the comparator arms, the calculated absolute effect was 89 more per 1000 patients surviving at 12 months (95% CI 63 more to 112 more). Pooled HR for PD-1/PD-L1 inhibitors was 0.74, 95% CI 0.68–0.80, while the pooled HR for CTLA-4 inhibitors was 0.92, 95% CI 0.81–1.06. The between-study heterogeneity was high ($I^2$ 87.2%) and prespecified subgroup analysis demonstrated a significant differential effect between

PD-1/PD-L1 inhibitors and CTLA-4 inhibitors. There is a high certainty of evidence that chemo-ICI results in a moderate improvement in OS in first-line ES-SCLC.

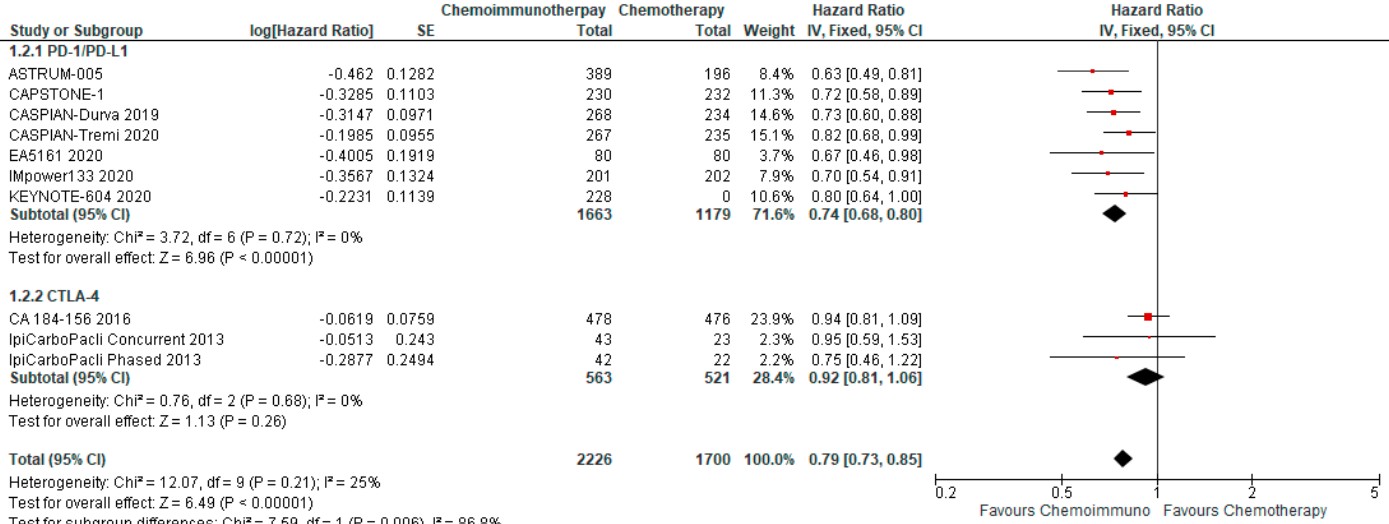

**Figure 2.** Forest Plot of hazard ratios for overall survival of patients with ES-SCLC and treated with chemo-ICI versus standard chemotherapy. PD–1/PD–L1, programmed death (ligand)–1; CTLA–4, cytotoxic T-lymphocyte-associated protein 4; HR, hazard ratio; CI, confidence interval; IV, inverse variance.

### 3.4.3. ORR

ORR was available from all included studies. The median ORR was 62% (range 32.5–80.2%) in all treatment groups and 61.9% (range 47–70.4%) in the control groups. There was no significant increase in ORR for patients randomised to chemo-ICI compared with chemotherapy alone (risk ratio 1.06, 95% CI 1.01–1.11) (Figure 3). There was limited heterogeneity ($I^2$ = 16%) and subgroup analysis did not reveal a significant difference in response rates between PD-1/PD-L1 and CTLA-4 inhibitors.

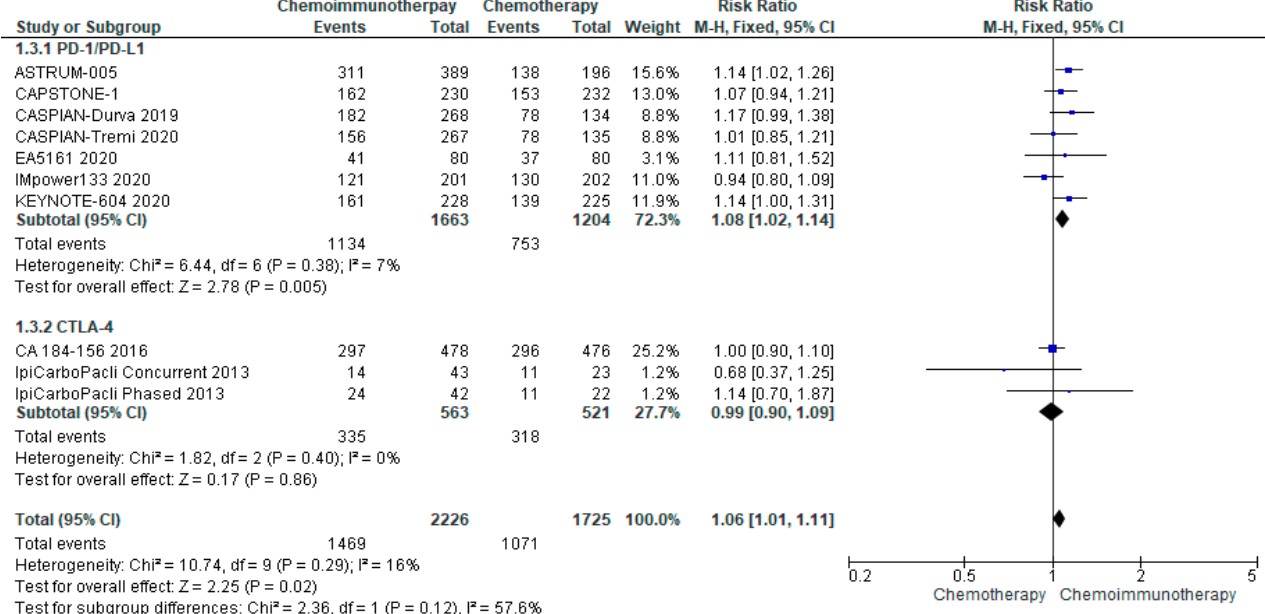

**Figure 3.** Forest Plot of risk ratios for ORR in patients with ES-SCLC and treated with chemo-ICI versus standard chemotherapy. PD–1/PD–L1, programmed death (ligand)–1; CTLA–4, cytotoxic T-lymphocyte-associated protein 4; HR, hazard ratio; CI, confidence interval; M-H, Mantel–Haenszel.

3.4.4. Duration of Response

DoR, available from seven studies and eight treatment arms, was comparable between treatment and control groups with a median of 5.15 months (range 4.01–5.6 months) in the treatment group as compared with a median of 3.7 months (range 3.3–5.1 months) in the control groups. The pooled DoR demonstrated a mean difference of 0.13 months (95% CI −0.08–0.35) in favour of chemo-ICI (Figure 4). There was no evidence of heterogeneity ($I^2$ = 5%).

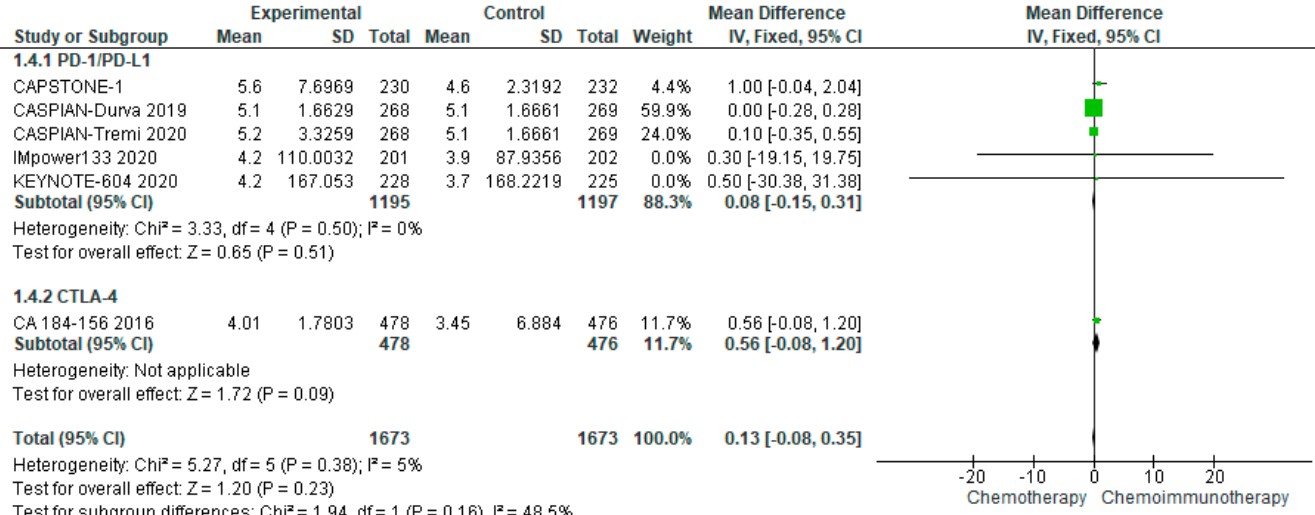

**Figure 4.** Forest Plot of mean difference in DoR in patients with ES-SCLC and treated with chemo-ICI versus standard chemotherapy. PD–1/PD–L1, programmed death (ligand)–1; CTLA–4, cytotoxic T-lymphocyte-associated protein 4; HR, hazard ratio; CI, confidence interval; IV, inverse variance.

*3.5. Adverse Events*

3.5.1. All Grade Adverse Events

There was a small increase in the overall incidence of all grade adverse events between chemo-ICI and chemotherapy with a pooled risk ratio of 1.03, 95% CI 1.01–1.06 (Figure 5). There was a risk difference of 26 more patients per 1000 (95% CI, 9 more to 53 more) who would experience all grade toxicities with chemo-ICI. There is moderate certainty of the evidence of a small increased risk of all-grade toxicity with chemo-ICI as compared with chemotherapy. There was significant heterogeneity ($I^2$ = 77%, $p$ = 0.0002), that was attenuated with the pre-specified subgroup analysis of PD-1/PD-L1 inhibitors versus CTLA-4 inhibitors. With PD-1/PD-L1 inhibitors, the pooled risk ratio was 1.02 (95% CI, 1.00–1.03), and there was limited heterogeneity with $I^2$ = 21%. The absolute risk difference was approximately 19 more per 1000 patients (95% CI, 0 fewer to 29 more) who would experience all grade toxicities with chemo-ICI. In the subgroup of CTLA-4 inhibitors, the pooled risk ratio was 1.07 (95% CI, 1.00–1.13) with limited heterogeneity ($I^2$ = 24%). The risk difference was 54 more patients per 1000 (0 fewer to 100 more) experiencing all grade toxicities with chemo-ICI with a CTLA-4 inhibitor. Both subgroups had high certainty of the evidence of a small increase in all-grade toxicity with chemo-ICI.

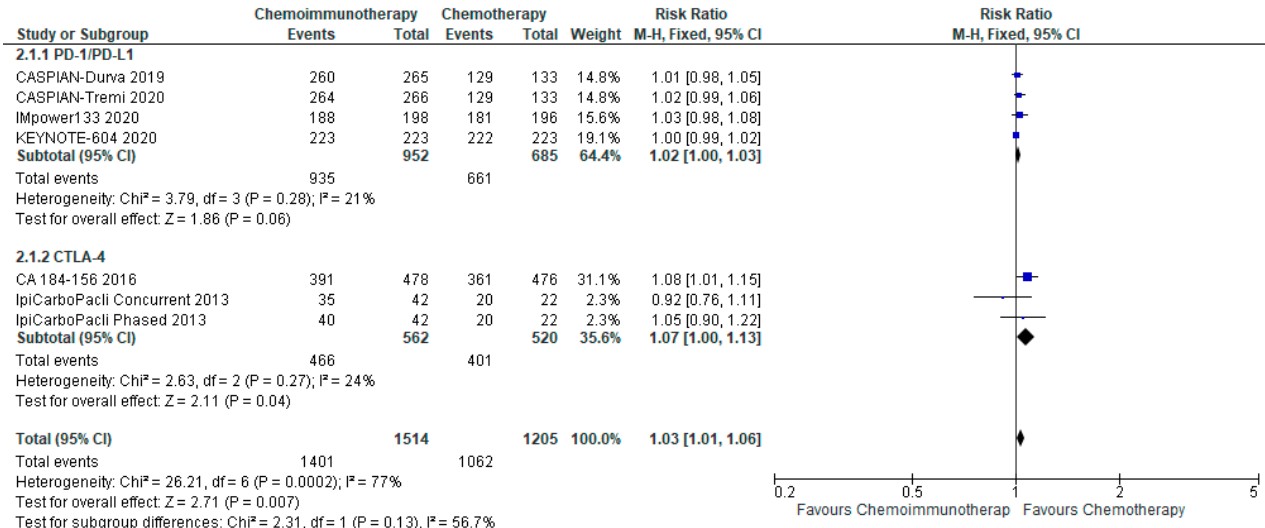

**Figure 5.** Forest plot of risk ratios for all grade adverse events in patients with ES-SCLC and treated with chemo-ICI versus standard chemotherapy. PD-1/PD-L1, programmed death (ligand)-1; CTLA-4, cytotoxic T-lymphocyte-associated protein 4; HR, hazard ratio; CI, confidence interval; M-H, Mantel–Haenszel.

### 3.5.2. Grade 3–4 Adverse Events

There was no significant increase in grade 3–4 adverse events with a pooled risk ratio of 1.06 (95% CI, 1.00–1.12) with chemo-ICI and no evidence of heterogeneity (Figure 6). The absolute risk difference was 33 more patients per 1000 (95% CI, 0 fewer to 67 more) experiencing grade 3–4 toxicity with chemo-ICI with moderate certainty of evidence.

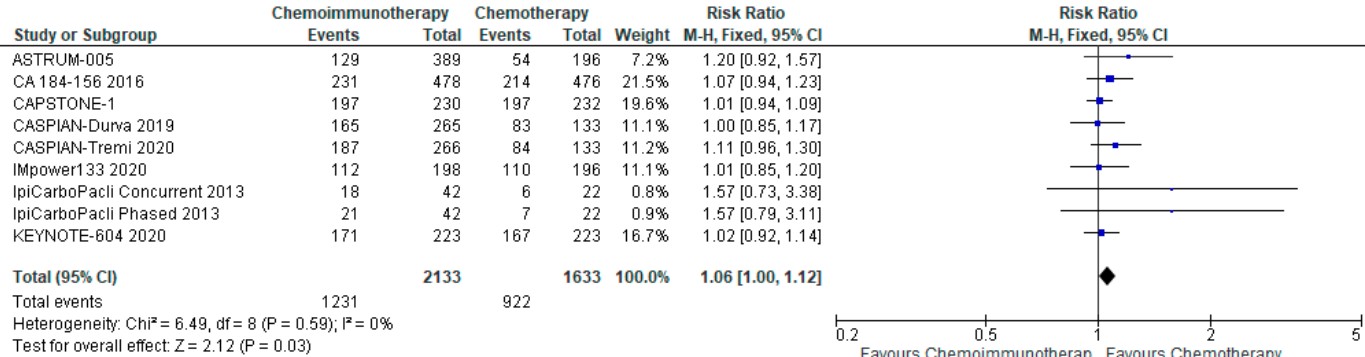

**Figure 6.** Forest plot of risk ratios for Grade 3–4 adverse events in patients with ES-SCLC and treated with chemo-ICI versus standard chemotherapy. PD-1/PD-L1, programmed death (ligand)-1; CTLA-4, cytotoxic T-lymphocyte-associated protein 4; HR, hazard ratio; CI, confidence interval; M-H, Mantel-Haenszel.

### 3.5.3. Immune-Related Adverse Events

The iRAEs were reported separately in 3 studies and 4 treatment arms. The rate of Grade 1–4 iRAE ranged from 20% to 38.9%. The most common iRAEs with a >10% incidence were rash and hypothyroidism, and all other iRAE incidences were <10%. Grade 3–4 iRAEs were only reported for 2 studies and ranged from 5% to 7.2% (Table 2). Three trials reported on colitis, pneumonitis, and hepatitis, while only 2 studies reported on rash and nephritis. No dedicated iRAE data were available from studies with CTLA-4 inhibitor backbones.

**Table 2.** Immune-related adverse events. Data from CASPIAN D, D+T (not available), IMpower133, and KEYNOTE-604. CA-184-156. PD-1/PD-L1, programmed death (ligand)-1.

| | Overall | |
|---|---|---|
| **irAE (3 PDL1)** | **Range (%)** | **Median** |
| Grade 1–4 * | 20–38.9 | 30.4 |
| Grade 3–4 | 5–7.2 | 6.1 |
| Grade 1–4 Colitis | 1.3–2 | 1.5 |
| Grade 3–4 Colitis | 0.4–1 | 0.4 |
| Grade 1–4 Rash | 2–18.7 | 10.35 |
| Grade 3–4 Rash | 0–2 | 0 |
| Grade 1–4 Pneumonitis | 2–4 | 3 |
| Grade 3–4 Pneumonitis | 0.5–1.3 | 1 |
| Grade 1–4 Hepatitis | 1.8–7.1 | 3 |
| Grade 3–4 Hepatitis | 1.3–6 | 1.5 |
| Grade 1–4 Nephritis | 0.5–0.9 | 0.7 |
| Grade 3–4 Nephritis | 0.4–0.5 | 0.45 |

### 3.6. Quality of Life

QoL was measured in CASPIAN and IMpower133 with validated tools (EORTC QLQ C30 and EORTC QLQ LC13). A pooled analysis was not thereby necessitating a narrative summary. PRO endpoints were pre-specified with QLQ-C30 v3 looking at fatigue and appetite loss, and with QLQ-LC13 including cough, dyspnoea, and chest pain. Higher scores for symptom items indicate greater symptom severity, while higher scores for function and global health status items indicate better function and health status. Both trials demonstrated comparable baselines between both arms amongst all domains in both questionnaires (Table 3). Both studies found similar improvements or maintenance of pre-treatment function in the two arms but with earlier tapering of effect in the comparator group. IMpower133 showed initial improvements in HRQoL in the chemo-ICI group persisting through to week 54, while the comparator group showed deterioration in initial benefits at week 21. CASPIAN also demonstrated improvement in symptom burden in both groups with treatment though the adjusted mean change in appetite loss was significantly greater with chemo-ICI (adjusted mean change from baseline: −12.7 vs. −8.2; estimated difference: −4.5; 95% CI, −9.04 to −0.04, nominal $p = 0.009$). Both studies also reported PROs as the time to deterioration (TTD) defined as the time from randomisation to the first clinically meaningful deterioration (≥10-point increase from baseline for symptoms; ≥10-point decrease from baseline for function and global health status). CASPIAN reported longer median TTD in the chemo-ICI arm for all function and symptom scales and a significant improvement in TTD in the domains of cognitive functioning, emotional functioning, physical functioning, role functioning, social functioning, appetite loss, constipation, diarrhoea, dyspnoea and insomnia, haemoptysis, arm/shoulder pain, and other pain. IMpower133 reported TTD as similar between both arms with significant differences identified in favour of chemo-ICI only for dyspnoea (HR 0.75; 95% CI, 0.55–1.02). There was a trend towards improvement in the remaining domains with chemo-ICI but no significant differences between the two arms.

**Table 3.** Baseline HRQoL Scores from CASPIAN and IMpower133. D+EP, durvalumab and etoposide platinum; EP, etoposide platinum; A+EP, atezolizumab and etoposide platinum.

| | CASPIAN | | IMpower133 | |
|---|---|---|---|---|
| | **D+EP** | **EP** | **A+EP** | **EP** |
| Global Health Status | 56.0 | 54.1 | 51.6 | 53.7 |
| Physical Functioning | 72.2 | 70.7 | 70.7 | 71.9 |

**Table 3.** *Cont.*

| | CASPIAN | | IMpower133 | |
|---|---|---|---|---|
| | **D+EP** | **EP** | **A+EP** | **EP** |
| Pain | 28.4 | 29.5 | 33.6 | 31.9 |
| Appetite Loss | 24.2 | 25.6 | 28.9 | 27.4 |
| Cough | 41.5 | 40.5 | 42.2 | 42.9 |
| Dyspnoea | 36.5 | 38.5 | 34.3 | 29.6 |
| Fatigue | 35.3 | 37.1 | 42.0 | 38.7 |
| Insomnia | 29.7 | 33.9 | 37.6 | 34.1 |

## 4. Discussion

The pooled analysis of eight trials evaluating combined chemo-ICI compared with standard-of-care chemotherapy in the first-line treatment of ES-SCLC demonstrates a significant improvement in PFS and OS. The relative risks of death or progression were both reduced by 20–25%. The results of this meta-analysis demonstrated a significant difference in outcomes between CTLA-4 and PD-1/PD-L1 inhibitors, with decreased benefit with CTLA-4 inhibitors. The CASPIAN trial showed no incremental value to the addition of tremelimumab to durvalumab plus chemotherapy and the ipilimumab trials failed to show significant improvements in efficacy and survival. Current data do not support the use of CTLA-4 inhibitors in the setting of ES-SCLC.

There was no difference in ORR or evidence of heterogeneity between PD-1/PD-L1 and CTLA-4 inhibitors. Historically, ORR is higher for combined CTLA-4 and PD-1 inhibitor therapy than for either single agent CTLA-4 (i.e., ipilimumab), or PD1-inhibitors (i.e., nivolumab or pembrolizumab) in a melanoma population, [20–23]. In NSCLC, chemo-ICI has demonstrated improved ORR [10,11,24] along with significant survival benefits. However, this trend of improved ORR was not observed in the ES-SCLC setting with median ORR only slightly improved with PD-1/PD-L1 inhibitors, but otherwise similar between the CTLA-4 inhibitor and combination ICI arms. Overall, there is no indication of a significant increase in ORR with the addition of ICI to the established response rates with standard-of-care chemotherapy in SCLC. Similarly, there was no evidence that DoR was improved with the addition of ICI to chemotherapy. This discrepancy between response to therapy and survival benefit suggests that ORR and median DoR may not be robust surrogate endpoints for PFS and OS for ICI therapy in SCLC patients [33,34]. Partly this may be due to the inability of these outcomes to adequately differentiate the subgroups of patients with ICI who achieve durable stable disease as witnessed by the OS data [35]. The OS benefit with chemo-ICI was initially observed at 4–6 months, with gradual separation of the survival curves beyond that time point suggesting a small proportion of patients achieving longer-term survival with chemo-ICI. This has been corroborated by the characterisation of long-term survivors in both the IMpower133 and Keynote-604 trials [36,37]. Therefore, ORR may act as a surrogate marker for the initial benefit of chemotherapy but may not reflect the prolonged and delayed benefit attributed to ICI. Further follow-up of the data is required.

Identification of subgroups of patients with greater benefit from ICI in SCLC will be key to maximising survival benefits and potentially identifying patients who will derive ongoing durable responses. In NSCLC, PD-L1 and TMB are considered predictive biomarkers and identify a subgroup of patients who are likely to benefit from ICI. However, these biomarkers are not predictive of efficacy from ICI in the setting of SCLC on subgroup analyses. In an exploratory analysis, IMpower133 explored a blood-based mutational burden assay as a predictive marker. There was no survival benefit based on TMB status [36]. In Checkmate032, a study assessing nivolumab alone or nivolumab and ipilimumab in recurrent ES-SCLC, both PD-L1 status and TMB were assessed as predictive biomarkers with no evidence of an association between PD-L1 expression and ORR [12]. In addition to biomarkers, there have been early investigations into identifying SCLC subtypes based on transcription factor expression [38], as well as varying immune signatures [39]. However,

these models are not yet validated [40]. Overall, predictive biomarkers continue to be elusive, and further data are required to appropriately select patients who may derive a durable benefit with chemo-ICI above the expected benefits of chemotherapy in ES-SCLC.

ES-SCLC treatment is palliative in intent and toxicity is an important consideration for both clinicians and patients. Combined chemo-ICI offers a modest improvement in OS with a comparable toxicity profile to chemotherapy alone with a slight increase with chemo-ICI in Grade 1–4 adverse events. This is similar to the data from the advanced NSCLC population with combined chemo-ICI resulting in comparable toxicity rates [10,11]. The toxicity data had significant heterogeneity that was attenuated by the pre-specified subgroup analysis given previous data supporting increased toxicity with CTLA-4 inhibitors as compared with PD-1/PD-L1 inhibitors [41,42]. The persistent heterogeneity may partially be explained by the decision to include the CASPIAN CTLA-4 inhibitor arm under the PD-1/PD-L1 subgroup. However, when assessing Grade 3–4 AE, there appears to be a comparable toxicity profile between the two arms with no evidence of heterogeneity. This suggests that the toxicity profile may be driven by low-grade toxicity that may have less clinical importance. Heterogeneity may also partially be driven by the different chemotherapy backbones used (cisplatin vs. carboplatin and etoposide vs. paclitaxel) as iRAEs were captured separately. Specific toxicities including laboratory derangements (anemia, thrombocytopenia, neutropenia), febrile neutropenia, and fatigue are reported separately.

HRQoL is also an important consideration in palliative intent treatment and value is placed on incremental benefits in survival if they do not come at expense of quality of life. Patients with SCLC have worse HRQoL and PROs compared with the general population, with untreated ES-SCLC having the greatest impact on HRQoL [43]. Unfortunately, only two studies assessed HRQoL formally with pre-specified secondary outcomes (IMpower133 and CASPIAN). Overall, there was no demonstrated detriment to QoL with the addition of ICI to chemotherapy. Standardisation in the reporting of HRQoL is important to determine the impact of chemo-ICI in this patient population.

The trials were predominantly homogenous in the chemotherapy backbone and the administration schedule used in both arms. The majority of trials utilised the current standard of treatment of EP [44], however, the initial ipilimumab trial combined carboplatin and paclitaxel in SCLC which does have proven efficacy but is not used widely. Most trials also administered 4 cycles of platinum-etoposide consistently in both arms. CASPIAN was the only trial to have a differing treatment plan with the treatment arm receiving 4 cycles of EP with durvalumab and the comparator arm having up to 6 cycles of EP. Maintenance immunotherapy was offered following the completion of induction concurrent chemo-ICI in the majority of included trials.

The pooled results from these 8 trials must be reconciled with some methodological limitations. The primary limitation remains the heterogeneity of the treatment arms amongst the included trials. Despite the overall addition of ICI to chemotherapy being assessed, there was a mix of PD-1/PD-L1 inhibitors as well as CTLA-4 inhibitors. Despite PD-1 and PD-L1 inhibitors having a similar mechanism of action, there is a potential signal of improved efficacy with PD-1 inhibitors over PD-L1 inhibitors [45]. Conversely, there is an established difference in mechanism of action, efficacy, and toxicity between PD-1/PD-L1 inhibitors and CTLA-4 inhibitors. The variability within the treatment arms was recognised and a prespecified subgroup analysis revealed a significant difference in effect. However, a note must be made of the large, phase III trials evaluating PD-1/PD-L1 inhibitors versus the smaller, phase II trials evaluating CTLA-4 inhibitors. Additionally, CASPIAN was the only trial to assess the combination of dual ICI (PD-L1 and CTLA-4 inhibitors) with chemotherapy. This deviance was somewhat mitigated by grouping this arm with the PD-1/PD-L1 inhibitors so as not to overestimate the effect of CTLA-4 inhibitors. Finally, limited data were available for the EA5161 trial and the ASTRUM-005 trial and therefore ongoing assessment of the updated analysis will be required for an accurate assessment of the efficacy of the addition of ICI to chemotherapy in an ES-SCLC population.

### 5. Conclusions

Our review reports a pooled analysis of chemo-ICI versus chemotherapy in the first-line ES-SCLC setting and is the first to assess the differential efficacy of CTLA-4 inhibitors and PD-1/PD-L1 inhibitors. We also recognised the value of pooling additional clinical outcomes including ORR, DoR, and toxicity for clinical decision-making and assessment of risk. Finally, there is a paucity of QoL data that allow for quantitative pooling; however, our qualitative pooling provides further corroboration of the benefit from combined chemo-ICI. There have been limited advances in survival in ES-SCLC, and despite a modest improvement, chemo-ICI has pushed the needle forward in otherwise notoriously recalcitrant disease. The advancements in survival and efficacy come without significant impairment in HRQoL or increases in treatment-related toxicity, suggesting that the addition of ICI to SCLC treatment may continue to improve survival and alleviate symptom burden in this patient population. Further investigations into the accurate identification of predictive biomarkers will be of high yield to maximise the survival benefits of ICI in the setting of ES-SCLC.

**Author Contributions:** Conceptualisation, A.S. and P.M.E.; methodology, A.S., P.M.E. and G.R.P.; software, A.S.; validation, A.S., M.F., G.R.P. and P.M.E.; formal analysis, A.S.; data curation, A.S. and M.F.; writing—original draft preparation, A.S. and M.F.; writing—review and editing, A.S., M.F., G.R.P. and P.M.E.; visualisation, A.S.; supervision, G.R.P. and P.M.E. All authors have read and agreed to the published version of the manuscript.

**Funding:** This research received no external funding.

**Conflicts of Interest:** No declarations of interest for A.S. and M.F. G.R.P.—Honorarium (Takeda), Consulting Fees (Profound Medical, Merck, Astra-Zeneca), Close family member who is an employee of Roche Canada and who owns stock in Roche. P.M.E.—Honorarium (Takea, Pfizer, Merck, Bristol Meyers Squibb, Astra-Zeneca, Lilly, Jazz, Jansen, Novartis) No funding was received for this study.

### Appendix A

MEDLINE

| # | Searches | Results |
|---|---|---|
| 1 | small cell lung cancer.mp. | 66,316 |
| 2 | exp Small Cell Lung Carcinoma/ | 3993 |
| 3 | 1 or 2 | 67,529 |
| 4 | immunotherapy.mp. | 105,565 |
| 5 | Programmed Cell Death 1 Receptor/or PD1 inhibitor.mp. or PDL1 inhibitor.mp. | 6485 |
| 6 | CTLA-4 Inhibitor.mp. | 91 |
| 7 | pembrolizumab.mp. | 3958 |
| 8 | nivolumab.mp. or nivolumab/ | 5008 |
| 9 | ipilimumab.mp. or Ipilimumab/ | 3565 |
| 10 | avelumab.mp. | 422 |
| 11 | tremelimumab.mp. | 283 |
| 12 | atezolizumab.mp. | 1067 |
| 13 | durvalumab.mp. | 503 |
| 14 | 4 or 5 or 6 or 7 or 8 or 9 or 10 or 11 or 12 or 13 | 114,500 |
| 15 | Antineoplastic Agents/ | 277,204 |
| 16 | cisplatin.mp. or Cisplatin/ | 75,659 |
| 17 | carboplatin.mp. or Carboplatin/ | 17,588 |
| 18 | Paclitaxel.mp. or Paclitaxel/ | 37,346 |
| 19 | etoposide.mp. or Etoposide/ | 25,396 |
| 20 | chemotherapy.mp. | 441,677 |
| 21 | 15 or 16 or 17 or 18 or 19 or 20 | 669,235 |
| 22 | randomised controlled trial.pt. | 507,927 |
| 23 | controlled clinical trial.pt. | 93,720 |
| 24 | randomised.ab. | 483,103 |
| 25 | placebo.ab. | 208,619 |
| 26 | drug therapy.fs. | 2,212,406 |

| # | Searches | Results |
|---|----------|---------|
| 27 | randomly.ab. | 335,363 |
| 28 | trial.ab. | 509,238 |
| 29 | groups.ab. | 2,058,166 |
| 30 | 22 or 23 or 24 or 25 or 26 or 27 or 28 or 29 | 472,7391 |
| 31 | animals/ | 6,621,020 |
| 32 | human/ | 18,535,892 |
| 33 | 31 not (31 and 32) | 4,675,617 |
| 34 | 30 not 33 | 4,101,212 |
| 35 | 3 and 14 and 21 and 34 | 1125 |
| 35 | limit 34 to yr = "2010-Current" | 1040 |

## Appendix B

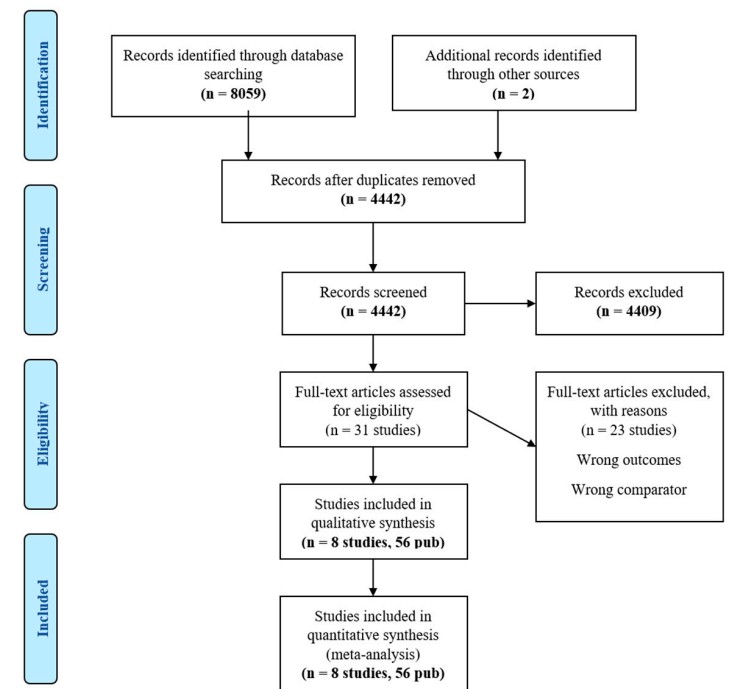

## Appendix C

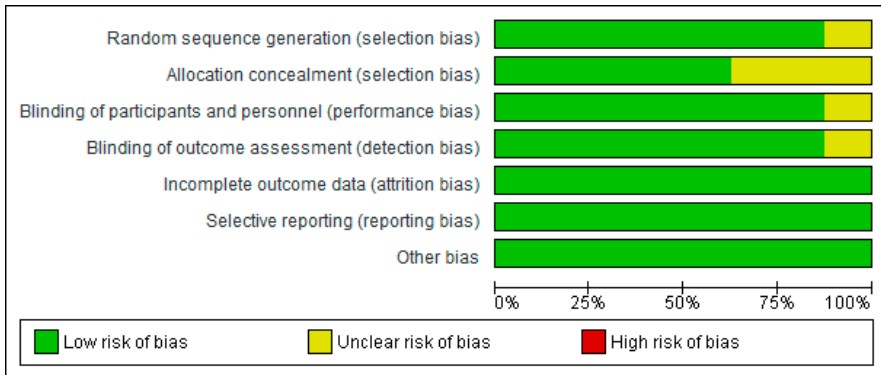

**Figure A1.** Risk of bias graph: review authors' judgements about each risk of bias item presented as percentages across all included studies.

## Appendix D

| Chemo-ICI compared with Chemotherapy in 1st line ES-SCLC | | | | | |
|---|---|---|---|---|---|
| Patient or population: 1st line ES-SCLC <br> Intervention: Chemo-ICI <br> Comparison: Chemotherapy | | | | | |
| Outcomes | № of Participants (Studies) Follow Up | Certainty of the Evidence (GRADE) | Relative Effect (95% CI) | Anticipated Absolute Effects | |
| | | | | Risk with Chemotherapy | Risk Difference with Chemo-ICI |
| Progression-Free Survival | 3952 (10 RCTs) | ⊕⊕⊕⊕ HIGH | HR 0.75 (0.70 to 0.80) [Survival] | 6 month | |
| | | | | 238 per 1000 | 103 more per 1000 (79 more to 128 more) |
| Overall Survival | 3952 (10 RCTs) | ⊕⊕⊕⊕ HIGH | HR 0.78 (0.73 to 0.84) [Survival] | 12 month | |
| | | | | 400 per 1000 | 89 more per 1000 (63 more to 112 more) |
| Objective Response Rate | 3952 (10 RCTs) | ⊕⊕⊕○ MODERATE [a] | RR 1.04 (0.98 to 1.10) | 601 per 1000 | 24 more per 1000 (12 fewer to 60 more) |
| Duration of Response | 2884 (6 RCTs) | ⊕⊕⊕○ MODERATE [a] | - | The median duration of Response was 3.7 months | MD 0.09 months higher (0.13 lower to 0.32 higher) |
| Grade 1–4 AE | 2719 (7 RCTs) | ⊕⊕⊕○ MODERATE [b] | RR 1.03 (1.01 to 1.06) | 881 per 1000 | 26 more per 1000 (9 more to 53 more) |
| Grade 1–4 AE: PD-1/PD-L1 | 1637 (4 RCTs) | ⊕⊕⊕⊕ HIGH | RR 1.02 (1.00 to 1.03) | 965 per 1000 | 19 more per 1000 (0 fewer to 29 more) |
| Grade 1–4 AE: CTLA-4 | 1082 (3 RCTs) | ⊕⊕⊕⊕ HIGH | RR 1.07 (1.00 to 1.13) | 771 per 1000 | 54 more per 1000 (0 fewer to 100 more) |
| Grade 3–4 AE | 2719 (9 RCTs) | ⊕⊕⊕○ MODERATE [a] | RR 1.06 (1.00 to 1.12) | 557 per 1000 | 33 more per 1000 (0 fewer to 72 more) |
| Grade 5 AE | 2736 (6 RCTs) | ⊕⊕○○ LOW [c] | RR 1.71 (0.90 to 3.25) | 12 per 1000 | 9 more per 1000 (1 fewer to 27 more) |
| Grade 1–4 Anemia | 1499 (5 RCTs) | ⊕⊕⊕○ MODERATE [a] | RR 0.98 (0.88 to 1.09) | 458 per 1000 | 9 fewer per 1000 (55 fewer to 41 more) |

| | | | | | |
|---|---|---|---|---|---|
| Grade 3–4 Anemia | 1644 (6 RCTs) | ⊕⊕○○ LOW [c] | RR 0.88 (0.69 to 1.12) | 149 per 1000 | 18 fewer per 1000 (46 fewer to 18 more) |
| Grade 1–4 Thrombocytopenia | 1499 (5 RCTs) | ⊕⊕○○ LOW [c] | RR 1.01 (0.85 to 1.21) | 218 per 1000 | 2 more per 1000 (33 fewer to 46 more) |
| Grade 3–4 Thrombocytopenia | 1688 (6 RCTs) | ⊕⊕○○ LOW [c] | RR 1.05 (0.79 to 1.41) | 93 per 1000 | 5 more per 1000 (19 fewer to 38 more) |
| Grade 1–4 Neutropenia | 1499 (5 RCTs) | ⊕⊕⊕○ MODERATE [a] | RR 1.01 (0.90 to 1.13) | 451 per 1000 | 5 more per 1000 (45 fewer to 59 more) |
| Grade 3–4 Neutropenia | 1688 (6 RCTs) | ⊕⊕⊕○ MODERATE [a] | RR 0.93 (0.81 to 1.07) | 313 per 1000 | 22 fewer per 1000 (60 fewer to 22 more) |
| Grade 3–4 Febrile Neutropenia | 1070 (3 RCTs) | ⊕⊕○○ LOW [c] | RR 0.74 (0.44 to 1.24) | 60 per 1000 | 16 fewer per 1000 (34 fewer to 14 more) |
| Grade 1–4 Fatigue | 1499 (5 RCTs) | ⊕⊕○○ LOW [c] | RR 1.06 (0.88 to 1.28) | 213 per 1000 | 13 more per 1000 (26 fewer to 60 more) |
| Grade 3–4 Fatigue | 1644 (6 RCTs) | ⊕⊕○○ LOW [c] | RR 1.68 (0.91 to 3.10) | 19 per 1000 | 13 more per 1000 (2 fewer to 39 more) |
| Grade 1–4 Nausea | 1521 (5 RCTs) | ⊕⊕⊕○ MODERATE [a] | RR 1.01 (0.88 to 1.16) | 337 per 1000 | 3 more per 1000 (40 fewer to 54 more) |
| Grade 3–4 Nausea | 1543 (5 RCTs) | ⊕⊕○○ LOW [c] | RR 0.42 (0.15 to 1.19) | 14 per 1000 | 8 fewer per 1000 (12 fewer to 3 more) |
| Grade 1–4 Vomiting | 1371 (3 RCTs) | ⊕⊕○○ LOW [c] | RR 0.96 (0.75 to 1.23) | 155 per 1000 | 6 fewer per 1000 (39 fewer to 36 more) |
| Grade 3–4 Vomiting | 1374 (3 RCTs) | ⊕⊕○○ LOW [c] | RR 0.42 (0.14 to 1.28) | 15 per 1000 | 8 fewer per 1000 (13 fewer to 4 more) |
| HRQoL | (2 RCTs) | - | | | |

The risk in the intervention group (and its 95% confidence interval) is based on the assumed risk in the comparison group and the relative effect of the intervention (and its 95% CI).
CI: Confidence interval; HR: Hazard Ratio; RR: Risk ratio; MD: Mean difference

GRADE Working Group grades of evidence
High certainty: We are very confident that the true effect lies close to that of the estimate of the effect
Moderate certainty: We are moderately confident in the effect estimate: The true effect is likely to be close to the estimate of the effect, but there is a possibility that it is substantially different
Low certainty: Our confidence in the effect estimate is limited: The true effect may be substantially different from the estimate of the effect
Very low certainty: We have very little confidence in the effect estimate: The true effect is likely to be substantially different from the estimate of effect

*Explanations*

a.    Effect cross null (1)
b.    $I^2 = 77\%$, p = 0.0002
c.    Events < 300 in dichotomous outcome, effect cross null

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
