# Peer review of "Chemo-Immunotherapy in First Line Extensive Stage Small Cell Lung Cancer (ES-SCLC): A Systematic Review and Meta-Analysis"

_curroncol, doi:10.3390/curroncol29120709_

Round 1

Reviewer 1 Report

This manuscript reports a systematic review and meta-analysis of chemoimmunotherapy for first-line extensive-stage small-cell lung cancer. Due to the relatively large number of reports in this area, this manuscript does not offer innovative recommendations. Therefore, it is premature to publish this manuscript in its current state.

Author Response

Thank you for taking the time to review our work and for your comments, we have reviewed 2 of the other recently published reviews and feel that our work adds valuable insight.

Other published reviews have focused on the benefits with PD-1/PD-L1 inhibitors, and rarely included CTLA-4 data. Our review reports a pooled analysis of chemo-ICI versus chemotherapy in the first line ES-SCLC setting and is the first to assess the differential efficacy of CTLA-4 inhibitors and PD-1/PD-L1 inhibitors with a prespecified sub-group analysis.

The majority of reviews focused on survival outcomes (PFS and OS), we also recognized the value of pooling additional clinical outcomes including ORR, DoR and toxicity for clinical decision making and assessment of risk. We also reported on toxicity (quantitative pooling of data) and HRQoL (qualitative pooling of data).  Toxicity and HRQoL are often overlooked in reviews and provide valuable information for clinicians to provide patient-centered care. 

Reviewer 2 Report

The SCLC is devastating and aggressive disease with high metastatic potential, and although the tumors initially respond to standard chemotherapy +/- radiotherapy, the disease relapses with deadly consequences due to chemo-resistance. Thus, survival rate of ES-SCLC patients is very low (e.g. PFS is shorter then six months, while median OS is approximately 10 months) and many efforts should be made in improving therapy regimen/options that will consequently provide survival improvements of SCLC patients. The manuscript „Chemo-immunotherapy in first line extensive stage small cell lung cancer (ES-SCLC): A systematic review and meta-analysis“ by Sathiyapalan et al. gives a systemic review and pooled analysis of eight trials with the aim to evaluate the clinical benefits and outcomes of combined ICI-chemo compared to standard chemotherapy in the first line treatment of ES-SCLC. Authors assessed several clinical parameters, such as survival data (PFS, OS, objective response rate (ORR), duration of response (DoR)), toxicity and health-related quality of life (HRQoL) and reported improvement in PFS and OS as well as reduced relative risk of death or disease progression with combined chemo-ICI. Interestingly, meta-analysis reported different outcomes of PD-1/PD-L1 and CTLA4, where latter showed reduced benefit. The topic of the manuscript is of great importance and the manuscript has potential, however several remarks need to be addressed and should be considered before publication.

There have been studies that performed systemic reviews and meta-analysis of clinical trials with the aim to investigate the clinical benefits of including ICI to first line chemotherapy in ES-SCLC (e.g. PMID: 32947924, PMID: 35032007). Consequently, I would not agree with the authors that their manuscript is the first such report (line 226 page 12), thus please rephrase. Could authors please explain and outline in the manuscript major findings that are not already addressed in two mentioned studies?

Please re-check the references 17, 18, 19 and 25 as they do not seem to fit the text at the pages 3 and 4, lines 97-98, 102-104, 119 and 139-140, respectively. Additionally, please check the reference 33 that does not seem to be in agreement with the text at page 6, lines 5-6 (incorrectly assigned designation of lines after inserting Table 1). Please replace the mentioned references accordingly.

In line 154 page 11, authors refer to the reference 39 that relates to the NSCLC Keynote-10 trial, but in the text they mention the SCLC Keynote-604 trial, so please add the adequate reference that refers to the SCLC trial

Minor remarks:

Please check the typos, e.g. line 9 page 1 – replace „treatmnt“ with „treatment”. Also, please replace “Impower133” with “IMpower133” throughout the text

Please add explanation of IV or M-H (e.g. IV, Inverse variance or M-H, Mantel-Haenszel) in Figure descriptions, accordingly

Detailed descriptions of Figures 5 and 6 are missing

Please add and remove point (“.”) in lines 145 and 152 (page 10), e.g. “chemotherapy This” and “point. suggesting”, respectively

Lines 203-206, please rephrase the sentence to make it more clear to the reader

Appendix C contains designation „Figure 5“ and description (please check lines 255-256 (page 14)), so please revise accordingly

References 14 and 29 are identical

Author Response

Thank you for taking the time to review our work and for your comments, we have addressed your suggestions as follows: 

There have been studies that performed systemic reviews and meta-analysis of clinical trials with the aim to investigate the clinical benefits of including ICI to first line chemotherapy in ES-SCLC (e.g. PMID: 32947924, PMID: 35032007). Consequently, I would not agree with the authors that their manuscript is the first such report (line 226 page 12), thus please rephrase. Could authors please explain and outline in the manuscript major findings that are not already addressed in two mentioned studies? We have rephrased the line stating our review to be the first. Upon review of the 2 other identified reviews, we have outlined the major differences and additional clinically relevant outcomes we reported including toxicity and HRQoL (in the conclusion). Toxicity and HRQoL are often overlooked in reviews and provide valuable information for clinicians to provide patient-centered care. 

Please re-check the references 17, 18, 19 and 25 as they do not seem to fit the text at the pages 3 and 4, lines 97-98, 102-104, 119 and 139-140, respectively. Additionally, please check the reference 33 that does not seem to be in agreement with the text at page 6, lines 5-6 (incorrectly assigned designation of lines after inserting Table 1). Please replace the mentioned references accordingly. In line 154 page 11, authors refer to the reference 39 that relates to the NSCLC Keynote-10 trial, but in the text they mention the SCLC Keynote-604 trial, so please add the adequate reference that refers to the SCLC trialThank you, we apologize for the oversight. These have been corrected now. 

Minor remarks: All of these have been corrected or addressed in the text. 

Please check the typos, e.g. line 9 page 1 – replace „treatmnt“ with „treatment”. Also, please replace “Impower133” with “IMpower133” throughout the text

Please add explanation of IV or M-H (e.g. IV, Inverse variance or M-H, Mantel-Haenszel) in Figure descriptions, accordingly

Detailed descriptions of Figures 5 and 6 are missing

Please add and remove point (“.”) in lines 145 and 152 (page 10), e.g. “chemotherapy This” and “point. suggesting”, respectively

Lines 203-206, please rephrase the sentence to make it more clear to the reader

Appendix C contains designation „Figure 5“ and description (please check lines 255-256 (page 14)), so please revise accordingly

References 14 and 29 are identical

Reviewer 3 Report

Authors conducted a systematic review and meta-analysis with mainly previous 8 studies. Outcomes such as PFS, OS, ORR, DoR and HRQoL were reviewed. Authors summarized huge amount of data that collected so far. I recommend the authors to improve the manuscript by the following.

Please check and correct the numbers carefully.

For example, on Table 1, Cheng (2022) ASTRUM-005, other manuscript (JAMA) showed 0.48 (HR for PFS), 5.7 (MPFS treatment).

Reck (2013) showed 12.94 for mOS (treatment).

Horn (2018) showed 0.70 (HR for OS)

Paz-Ares 2019 showed 0.73 (HR for OS) and 13.0 (mOS treatment), 10.3 (mOS control). etc

I recommend to add each reference number on Table1.

Correct some of the references, such as Ref. 27

Describe legends in detail for Fig. 5, Fig. 6 and Tables, especially the abbreviations clearly. 

Check the typos such as CAPSIAN on line 178.

Please rewrite the figures 1-6 to make it look clear.

Authors identified initially 8061 studies and some publications. Add and organize all on the excel file in the supplementary data.

Add data on sex differences to improve the manuscript.

Authors should mention the following manuscript.

First-Line Treatment for Advanced SCLC: What Is Left Behind and Beyond Chemoimmunotherapy. Front. Med., 25 May 2022

Author Response

Thank you for taking the time to review our work and for your comments, we have addressed your suggestions as follows: 

Authors conducted a systematic review and meta-analysis with mainly previous 8 studies. Outcomes such as PFS, OS, ORR, DoR and HRQoL were reviewed. Authors summarized huge amount of data that collected so far. I recommend the authors to improve the manuscript by the following.

Please check and correct the numbers carefully. Thank you and we apologize for the oversight. The numbers have been corrected in the tables and updated elsewhere as well in the text. 

For example, on Table 1, Cheng (2022) ASTRUM-005, other manuscript (JAMA) showed 0.48 (HR for PFS), 5.7 (MPFS treatment).

Reck (2013) showed 12.94 for mOS (treatment).

Horn (2018) showed 0.70 (HR for OS)

Paz-Ares 2019 showed 0.73 (HR for OS) and 13.0 (mOS treatment), 10.3 (mOS control). etc

I recommend to add each reference number on Table1. This has been added to all studies in Table 1

Correct some of the references, such as Ref. 27 The references have been corrected, particularly there are two references for ASTRUM-005 (the initial abstract and the subsequent publication). 

Describe legends in detail for Fig. 5, Fig. 6 and Tables, especially the abbreviations clearly. Detailed legends similarly to the other figures have been added for Figure 5 and 6 and all tables. 

Check the typos such as CAPSIAN on line 178. Corrected.

Please rewrite the figures 1-6 to make it look clear. We have tried to make the figures clearer by enlarging them rather than keeping them in line with the text. Unfortunately, given the software used, we are unable to change the font size/type or remove columns. 

Authors identified initially 8061 studies and some publications. Add and organize all on the excel file in the supplementary data. We have uploaded the excel file as supplementary data

Add data on sex differences to improve the manuscript. Thank  you for this comment. We agree that this would provide more information however prespecified survival data and toxicity data for the overall population and therefore these outcomes were not collected based on sex during the initial data abstraction. 

Authors should mention the following manuscript.

First-Line Treatment for Advanced SCLC: What Is Left Behind and Beyond Chemoimmunotherapy. Front. Med., 25 May 2022 

Thank you for bring to attention this well written review. We have reviewed it in detail and referenced it in our discussion regarding the need for identification of SCLC subtypes. 

Reviewer 4 Report

In this systematic analysis and meta-analysis, the authors reported a pooled analysis of chemotherapy + immunotherapy vs chemotherapy for first-line therapy in the extensive stage / SCLC setting. The authors included 8 articles in their final analysis and found that addition of immunotherapy to chemotherapy in extensive stage SCLC was associated with 22% risk reduction in death and 25% risk reduction in disease progression. 

I thought the manuscript was well written with sound analysis.  The authors appropriately discussed the limitations of the study.  Clearly, tremendous amount of effort was put forth by the authors.  Congratulations to the authors on this work!!

Author Response

Thank you for taking the time to review our work and for your comments.

Round 2

Reviewer 1 Report

The authors have addressed my concerns.

Reviewer 2 Report

I acknowledge authors for having addressed all my comments and for improving their manuscript.

Reviewer 3 Report

Nothing.